# Drawing early-bird tickets: Towards more efficient training of deep networks

**Haoran You, Chaojian Li, Pengfei Xu, Yonggan Fu, Yue Wang, Richard G. Baraniuk & Yingyan Lin**[*]
Department of Electrical and Computer Engineering
Rice University
Houston, TX 77005, USA
`{hy34, cl114, px5, yf22, yw68, yingyan.lin, richb}@rice.edu`

**Xiaohan Chen & Zhangyang Wang**[*]
Department of Computer Science and Engineering
Texas A&M University
College Station, TX 77843, USA
`{chernxh, atlaswang}@tamu.edu`

## Abstract

(Frankle & Carbin, 2019) shows that there exist *winning tickets* (small but critical subnetworks) for dense, randomly initialized networks, that can be trained alone to achieve a comparable accuracy to the latter in a similar number of iterations. However, the identification of these winning tickets still requires the costly train-prune-retrain process, limiting their practical benefits. In this paper, we discover for the first time that the **winning tickets can be identified at a very early training stage**, which we term as **Early-Bird (EB)** tickets, via low-cost training schemes (e.g., early stopping and low-precision training) at large learning rates. Our finding on the existence of EB tickets is consistent with recently reported observations that the key connectivity patterns of neural networks emerge early. Furthermore, we propose a mask distance metric that can be used to identify EB tickets with a low computational overhead, without needing to know the true winning tickets that emerge after the full training. Finally, we leverage the existence of EB tickets and the proposed mask distance to develop efficient training methods, which are achieved by first identifying EB tickets via low-cost schemes, and then continuing to train merely the EB tickets towards the target accuracy. Experiments based on various deep networks and datasets validate: 1) the existence of EB tickets and the effectiveness of mask distance in efficiently identifying them; and 2) that the proposed efficient training via EB tickets can achieve up to $5.8\times \sim 10.7\times$ **energy savings** while maintaining comparable or even better accuracy as compared to the most competitive state-of-the-art training methods, demonstrating a promising and easily adopted method for tackling the often cost-prohibitive deep network training. Codes available at `https://github.com/RICE-EIC/Early-Bird-Tickets`

## 1 Introduction

The recent record-breaking predictive performance achieved by deep neural networks (DNNs) motivates a tremendously growing demand to bring DNN-powered intelligence into numerous applications (Xu et al., 2020). However, the excellent performance of modern DNNs comes at an often prohibitive training cost due to the required vast volume of training data and model parameters. As an illustrative example of the computational complexity of DNN training, one forward pass of the ResNet50 (He et al., 2016a) model requires 4 GFLOPs (FLOPs: floating point operations) of computations and training requires $10^{18}$ FLOPs, which takes 14 days on one state-of-the-art NVIDIA M40 GPU (You et al., 2018). As a result, training a state-of-the-art DNN model often demands considerable energy, along with the associated financial and environmental costs. For example, a recent report shows that training a single DNN can cost over $10K US dollars and emit as much carbon as five cars in their lifetimes (Strubell et al., 2019), limiting the rapid development of DNN innovations and raising various environmental concerns.

The recent trends of improving DNN efficiency mostly focus on compressing models and accelerating inference. An empirically adopted practice is the so-called *progressive pruning and training*

---

[*]Correspondence should be addressed to: Zhangyang Wang and Yingyan Lin.

routine, i.e., training a large model fully, pruning it, and then retraining the pruned model to restore the performance (the process can be iterated several rounds). While this has been a standard practice for model compression (Han et al., 2015), some recent efforts start empirically linking it to the potential of more efficient training. Notably, the latest series of works (Frankle & Carbin, 2019; Liu et al., 2018b) reveals that dense, randomly-initialized networks contain small subnetworks which can match the test accuracy of original networks when trained alone themselves. These subnetworks are called *winning tickets*. Despite their insightful findings, there remains to be a major **gap** between the winning ticket observation and the goal of more efficient training, since winning tickets were only identified by pruning unimportant connections **after** *fully training* a dense network.

This paper closes this gap by demonstrating the **Early-Bird (EB) tickets** phenomenon: **the winning tickets can be drawn very early in training and with aggressively low-cost training algorithms**. Through a range of experiments on different DNNs and datasets, we observe the consistent existence of EB tickets and the cheap costs needed to reliably draw them, and develop a novel *mask distance* metric to detect their emergence. After being identified, re-training those EB tickets (using standard training) leads to comparable or even better final accuracies, compared to either standard training, or re-training the "ground-truth" winning tickets drawn after full training as in (Frankle & Carbin, 2019). Our observations seem to coincide with the recent findings by (Achille et al., 2019; Li et al., 2019) about the two-stage optimization trajectory in training. Taking advantage of EB tickets, we propose an efficient DNN training scheme termed *EB Train*. To our best knowledge, this is the first step taken towards exploiting winning tickets for a realistic efficient training goal.

Our contribution can be summarized as follow:

1. We discover the Early-Bird (EB) tickets, and show that they 1) consistently exist across DNN models and datasets; 2) can emerge very early in training; and 3) stay robust under (and sometimes even favor) various aggressive and low-cost training schemes (in addition to early stopping), including large learning rates and low-precision training.

2. We propose a practical, easy-to-compute mask distance as an indicator to draw EB tickets without accessing the "ground-truth" winning tickets (drawn after full training), fixing a major paradox for connecting winning tickets with the efficient training goal.

3. We design a novel efficient training framework based on EB tickets (EB Train). Experiments in state-of-the-art benchmarks and models show that EB Train can achieve up to **5.8× ~ 10.7× energy savings**, while maintaining the same or even better accuracy, compared to training with the original winning tickets.

## 2 RELATED WORKS

**Winning Ticket Hypothesis.** The lottery ticket hypothesis (Frankle & Carbin, 2019) first points out that a small subnetwork, called the winning ticket, can be identified by pruning a fully trained dense network; when training it in isolation with the same weight initialization once assigned to the corresponding weights in the dense network, one can restore the comparable test accuracy to the dense network. However, finding winning tickets hinged on costly (iterative) pruning and re-training. (Morcos et al., 2019) studies the reuse of winning tickets, transferable across different datasets. (Zhou et al., 2019) discovers the existence of supermasks that can be applied to an untrained, randomly-initialized network. (Liu et al., 2018b) argues that the weight initialization might make less difference when trained with a large learning rate, while the searched connectivity is more of the winning ticket's core value. It also explores the usage of both unstructured and (more hardware-friendly) structured pruning and shows that both lead to the emergence of winning tickets.

Another related work (Lee et al., 2019) prunes a network at single-shot with one mini-batch, in which the irrelevant connections are identified by a connection sensitivity criterion. Comparing to (Frankle & Carbin, 2019), the authors show their method to be more efficient in finding the good subnetwork (not the winning ticket), although its re-training accuracy/efficiency is found to be inferior, compared to training the "ground truth" winning ticket.

**Other Relevant Observations in Training.** (Rahaman et al., 2019; Xu et al., 2019) argue that deep networks will first learn low-complexity (lower-frequency) functional components, before absorbing high-frequency features: the former being more robust to perturbations. An important hint can be found in (Achille et al., 2019): the early stage of training seems to first discover the important connections and the connectivity patterns between layers, which becomes relatively fixed in the

later training stage. That seems to imply that the critical sub-network (connectivity) can be identified independent of, and seemingly also ahead of, the (final best) weights. Finally, Li et al. (2019) demonstrates that training a deep network with a large initial learning rate helps the model focus on memorizing easier-to-fit, more generalizable pattern faster and better – a direct inspiration for us to try drawing EB tickets using large learning rates.

**Efficient Inference and Training.** Model compression has been extensively studied for lighter-weight inference. Popular means include pruning (Li et al., 2017; Liu et al., 2017; He et al., 2018; Wen et al., 2016; Luo et al., 2017; Liu et al., 2018a), weight factorization (Denton et al., 2014), weight sharing (Wu et al., 2018a), quantization (Hubara et al., 2017), dynamic inference (Wang et al., 2018b; 2019b; Shen et al., 2020), network architecture search (Zoph & Le, 2017), among many others (Wang et al., 2018c;d). On the other hand, the literature on efficient training appears to be much sparser. A handful of works (Goyal et al., 2017; Cho et al., 2017; You et al., 2018; Akiba et al., 2017; Jia et al., 2018; Gupta et al., 2015) focus on reducing the total training time in paralleled, communication-efficient distributed settings. In contrast, our goal is to shrink the total resource cost for in-situ, resource-constrained training, as (Wang et al., 2019a) advocated. (Banner et al., 2018; Wang et al., 2018a) presented low-precision training, which is aligned with our goal and can be incorporated into EB Train (see later).

## 3 Drawing Early-bird Tickets: Hypothesis and Experiments

We hypothesize that the **winning tickets can emerge at a very early training stage**, which we term as an Early-Bird (EB) ticket. Consider a dense, randomly-initialized network $f(x; \theta)$, $f$ reaches a minimum validation loss $f_{\text{loss}}$ at the $i$-th iteration with a test accuracy $f_{\text{acc}}$, when optimized with stochastic gradient descent (SGD) on a training set. In addition, consider subnetworks $f(x; m \odot \theta)$ with a mask $m \in \{0, 1\}$ indicates the pruned and unpruned connections in $f(x; \theta)$. When being optimized with SGD on the same training set, $f(x; m \odot \theta_t)$ reach a minimum validation loss $f'_{\text{loss}}$ with a test accuracy $f'_{\text{acc}}$, where $\theta_t$ denotes the weights at the $t$-th iteration of training. The EB tickets hypothesis articulates that there exists $m$ such that $f'_{\text{acc}} \approx f_{\text{acc}}$ (even $\geq$), i.e., same or better generalization, with $t \ll i$ (e.g., early stopping) and a sparse $m$ (i.e., much reduced parameters).

Section 3 addresses **three key questions** pertaining to the EB ticket hypothesis. We first show via an extensive set of experiments, that EB tickets can be observed across popular models and datasets (Section 3.1). We then try to be more aggressive to see if high-quality EB tickets still emerge under "cheaper" training (Section 3.2). We finally reveal that EB tickets can be identified using a novel *mask distance* between consecutive epochs, thus no full training needed (Section 3.3).

### 3.1 Do Early-bird Tickets Always Exist?

We perform ablation simulations using two representative deep models: VGG16 (Simonyan & Zisserman, 2014) and pre-activation residual networks-101 (PreResNet101) (He et al., 2016b), on two popular datasets: CIFAR-10 and CIFAR-100. For drawing the tickets, we adopt a standard training protocol (Liu et al., 2018b) for both CIFAR-10 and CIFAR-100: the training takes 160 epochs in total and the batch size of 256; the initial learning rate is set to 0.1, and is divided by 10 at the 80th and 120th epochs, respectively; the SGD solver is adopted with a momentum of 0.9 and a weight decay of $10^{-4}$. For retraining the tickets, we keep the same setting by default.

We follow the main idea of (Frankle & Carbin, 2019), but instead prune networks trained at much earlier points (before the accuracies reach their final top values), to see if reliable tickets can still be drawn. We adopt the same channel pruning in (Liu et al., 2017) for all experiments since it is hardware friendly and aligns with our end goal of efficient training (Wen et al., 2016). Figure 1 reports the accuracies achieved by re-training the tickets drawn from different early epochs. All results consistently endorse that there exist high-quality tickets, at as early as the 20th epoch (w.r.t. a total of 160 epochs), that can achieve strong re-training accuracies. Comparing among different pruning ratios $p$, it is not too surprising to see over-pruning (e.g., $p = 70\%$) makes drawing good tickets harder, indicating a balance that we need to calibrate between accuracy and training efficiency.

Two more striking observations from Figure 1: 1) there consistently exist EB tickets drawn at certain early epoch ranges, that outperform those drawn in a later stages, including the "ground-truth" winning tickets drawn at the 160th epoch. That intriguing phenomenon implies the possible "over-cooking" when networks try to identify connectivity patterns at later stage (Achille et al., 2019),

Figure 1: Retraining accuracy vs. epoch numbers at which the subnetworks are drawn, for both PreResNet101 and VGG16 on the CIFAR-10/100 datasets, where $p$ indicates the channel pruning ratio and the dashed line shows the accuracy of the corresponding dense model on the same dataset, ☆ denotes the retraining accuracies of subnetworks drawn from the epochs with the best search accuracies, and error bars show the minimum and maximum of three runs.

that might hamper generalization; 2) some EB tickets are able to outperform even their unpruned, fully-trained models (e.g., the dashlines), potentially thanks to the sparse regularization learned by EB tickets.

Table 1: The retraining accuracy of subnetworks drawn at different training epochs using different learning rate schedules, with a pruning ratio of 0.5. Here [0, 100] represents $[0_{LR \to 0.01}, 100_{LR \to 0.001}]$ while [80, 120] denotes $[80_{LR \to 0.01}, 120_{LR \to 0.001}]$, for compactness.

| | LR Schedule | Retrain acc. (%) (CIFAR-100) | | | | Retrain acc. (%) (CIFAR-10) | | | |
|---|---|---|---|---|---|---|---|---|---|
| | | 10 | 20 | 40 | final | 10 | 20 | 40 | final |
| VGG16 | [0, 100] | 66.70 | 67.15 | 66.96 | 69.72 | 92.88 | 93.03 | 92.80 | 92.64 |
| | [80, 120] | **71.11** | **71.07** | **69.14** | **69.74** | **93.26** | **93.34** | **93.20** | **92.96** |
| PreResNet101 | [0, 100] | 69.68 | 69.69 | 69.79 | 70.96 | 92.41 | 92.72 | 92.42 | 93.05 |
| | [80, 120] | **71.58** | **72.67** | **72.67** | **71.52** | **93.60** | **93.46** | **93.56** | **93.42** |

## 3.2 DO EARLY-BIRD TICKETS STILL EMERGE UNDER LOWER-COST TRAINING?

For EB tickets, only the important connections (connectivity) found by them matter, while the weights are to be re-trained anyway. One might hence come up with an idea, that more aggressive and "cheaper" training methods might be applicable to further shrink the cost of finding EB tickets (on top of the aforementioned early stopping of training thanks to the identification of EB tickets ), as long as the same significant connections still emerge. We experimentally investigate the impacts of two schemes (which are using appropriate large learning rates and training in lower precision): EB tickets are observed to survive well under both of them.

**Appropriate large learning rates are important to the emergence of EB tickets.** We first vary the learning rate schedule in the above experiments. The original schedule is denoted as $[80_{LR \to 0.01}, 120_{LR \to 0.001}]$, i.e., starting from 0.1, decayed to 0.01 at the 80th epoch, and further decayed to 0.001 at the 120th epoch. In comparison, we test a new learning rate schedule $[0_{LR \to 0.01}, 100_{LR \to 0.001}]$: starting from 0.01 (the 0th epoch), and decayed to 0.001 at the 100th epoch. After drawing tickets, we re-train them all using the same learning rate schedule $[80_{LR \to 0.01}, 120_{LR \to 0.001}]$ for sufficient training and a fair comparison. We can see from Table 1 that high-quality EB tickets always emerge earlier when searching with the schedule of a larger learning rates, i.e., $[80_{LR \to 0.01}, 120_{LR \to 0.001}]$, whose final accuracies are also better. Note that the earlier emergence of good EB tickets contributes to lowering the training costs. It was observed that larger learning rates are beneficial for training the dense model fully to draw the winning tickets in (Frankle & Carbin, 2019; Liu et al., 2018b): here we show this benefit extends to EB tickets too. More experiments with larger learning rates can be found in Appendix A.2.

**Low-precision training does not destroy EB tickets.** We next examine the emergence of EB tickets within a state-of-the-art low-precision training algorithm (Wu et al., 2018b) , where all model weights, activations, gradients and errors are quantized to 8 bits throughout training. Note that here we only apply low-precision training to the stage of identifying EB tickets, and afterwards the tickets are trained in the same full-precision as in Section 3.1. Figure 2 shows the retraining accuracy and total number of FLOPs (EB ticket search (8 bits) + retraining (32 bits floating points)) for VGG16 and CIFAR-10/100. We can see that EB tickets still emerge when using aggressively low-precision for identifying EB tickets: the general trends resemble the full-precision training cases in Figure 1, except the emergence of good EB tickets seem to become even earlier up to initial epochs. In this

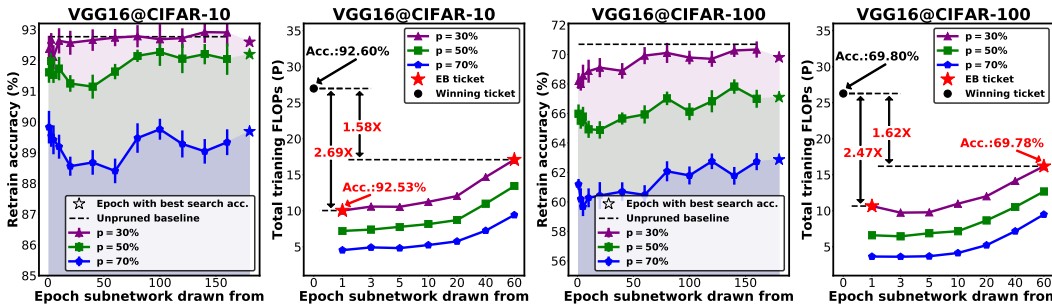

Figure 2: Retraining accuracy and total training FLOPs comparison vs. epoch number at which the subnetwork is drawn, when using 8 bits precision during the stage of identifying EB tickets based on the VGG16 model and CIFAR-10/100 datasets, where $p$ indicates the channel-wise pruning ratio and the dashed line shows the accuracy of the corresponding dense model on the same dataset.

way, it will lead to cost savings in finding EB tickets, since low-precision updates can aggressively save energy compared to their full-precision baseline, as shown in the Table 2.

### 3.3 HOW TO IDENTIFY EARLY-BIRD TICKETS PRACTICALLY?

**Distance between Ticket Masks.** For each time of pruning, we define a binary mask of the drawn ticket (pruned subnetwork) w.r.t. the full dense network. We follow (Liu et al., 2017) to consider the scaling factor $r$ in batch normalization (BN) layers as indicators of the corresponding channels' significance. Given a target pruning ratio $p$, we then prune the channels with top $p$-percentage smallest $r$ values. Denote the pruned channels as 0 while the kept ones as 1, the original network can be mapped into a binary "ticket mask". For any two sub-networks pruned from the same dense model, we calculate their *mask distance* via the Hamming distance between their two ticket masks.

**Detecting EB Tickets via Mask Distances.** We first visualize and observe the global behaviors of mask distances between consecutive epochs. Figure 3 plots the *pairwise mask distance matrices* ($160 \times 160$) of the VGG16 and PreResNet101 experiments on CIFAR-100 (from Section 3.1), at different pruning ratios $p$, where $(i, j)$-th element in a matrix denotes the mask distance between subnetworks drawn from the $i$-th and $j$-th epochs in that corresponding experiment (160 epochs in total for all). For the ease of visualization, all elements in a matrix are linearly normalized between 0 and 1; Therefore, in the resulting matrix, a lower value (close to 0) indicates a smaller mask distance and is highlighted with a warmer color (same hereinafter).

Figure 3 demonstrates fairly consistent behaviors. Taking VGG16 on CIFAR-100 ($p = 0.2$) for an example: 1) at the very beginning, the mask distances change rapidly between epochs, manifested by the quickly "cooling" colors (yellow → green), from the diagonal line (lowest since that is comparing an epoch's mask with itself), to off-diagonal (comparing different epochs; the more an element deviates from the diagonal, the more distant the two epochs are away from each other); 2) after ~10 epochs, the off-diagonal elements become "yellow" too, and the color transition becomes smoother from diagonal to off-diagonal, indicating the masks change mildly only after passing this point; 3)

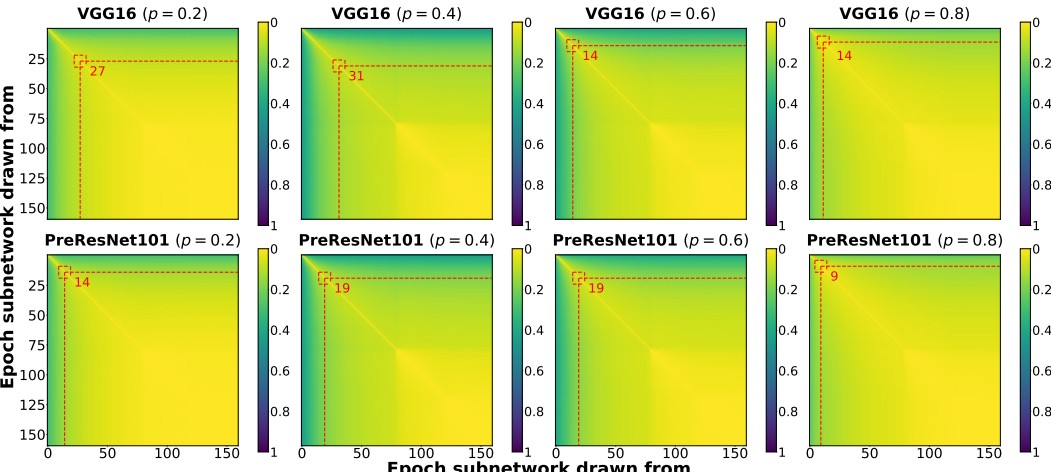

Figure 3: Visualization of the pairwise mask distance matrix for VGG16 and PreResNet101 on CIFAR-100.

after ~80 epochs, the mask distances almost remain unchanged across epochs. Similar trends are observed in other plots too. It seems to concur the hypothesis in (Achille et al., 2019) that a network first learns important connectivity patterns and then fixes them and further tune their weights.

Our observation that the ticket masks quickly become stable and hardly changed in early training stages supports drawing EB tickets. We therefore measure the mask distance between the consecutive epochs, and draw EB tickets when such distance is smaller than a threshold $\epsilon$. Practically, to improve the reliability of EB tickets, we will stop to draw EB tickets when the last five recorded mask distances are all smaller than $\epsilon$, to avoid some irregular fluctuation in early training stages. In Figure 3, the red lines indicate the identification of EB tickets when $\epsilon$ is set to 0.1.

## 4 EFFICIENT TRAINING VIA EARLY BIRD TICKETS

In this section, we present an efficient DNN training scheme, termed as EB Train, which leverages 1) the existence of EB tickets and 2) the proposed mask distance that can detect the emergence of EB tickets for time- and energy-efficient training. We will first provide a conceptual overview, and then describe the routine of EB Train, and finally show the evaluation performance of EB Train by benchmarking it with state-of-the-art methods of obtaining compressed DNN models based on representative datasets and DNNs.

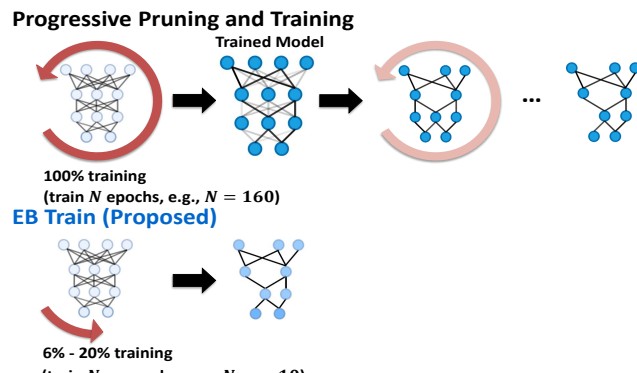

Figure 4: A high-level overview of the commonly adopted *progressive pruning and training* scheme and our EB Train.

### 4.1 WHY IS EB TRAIN MORE EFFICIENT?

**EB Train vs. *Progressive Pruning and Training*.** Figure 4 illustrates an overview of our proposed EB Train and the *progressive pruning and training* scheme, e.g., as in (Frankle & Carbin, 2019). In particular, the *progressive pruning and training* scheme adopts a three-step routine of 1) training a large and dense model, 2) pruning it, and 3) then retraining the pruned model to restore performance, and these three steps can be iterated (Han et al., 2015). The first step often dominates (e.g., occupy 75% training FLOPs when using the PreResNet101 model and CIFAR-10 dataset) in terms of training energy and time costs as it involves training a large and dense model. The key feature of our EB Train scheme is that it replaces the aforementioned steps 1 and 2 with a lower-cost step of detecting the EB tickets, i.e., enabling early stopping during the time- and energy-dominant step of training the large and dense model, thus promising large savings in both training time and energy. For example, assuming the first step of the *progressive pruning and training* scheme requires $N$ epochs to sufficiently train the model for achieving the target accuracy, the proposed EB Train needs only $N_{EB}$ epochs to identify the EB tickets by making use of the mask distance that can detect the emergence of EB tickets, where $N_{EB} \ll N$ (e.g., $N_{EB}/N = 12.5\%$ in the experiments summarized in Figure 1 and $N_{EB}/N = 6.25\%$ in the experiment summarized in Table 1).

---

**Algorithm 1:** The Algorithm for Searching EB Tickets

---
1: Initialize the weights $\boldsymbol{W}$, scaling factor $r$, pruning ratio $p$, and the FIFO queue $Q$ with length $l$;
2: **while** $t$ (epoch) $< t_{max}$ **do**
3:     Update $\boldsymbol{W}$ and $r$ using SGD training;
4:     Perform structured pruning based on $r_t$ towards the target ratio $p$;
5:     Calculate the **mask distance** between the current and last subnetworks and add to $Q$.
6:     $t = t + 1$
7:     **if** $\textbf{Max}(Q) < \epsilon$ **then**
8:         $t^* = t$
9:         **Return** $f(x; m_{t^*} \odot \boldsymbol{W})$ (EB ticket);
10:     **end if**
11: **end while**

---

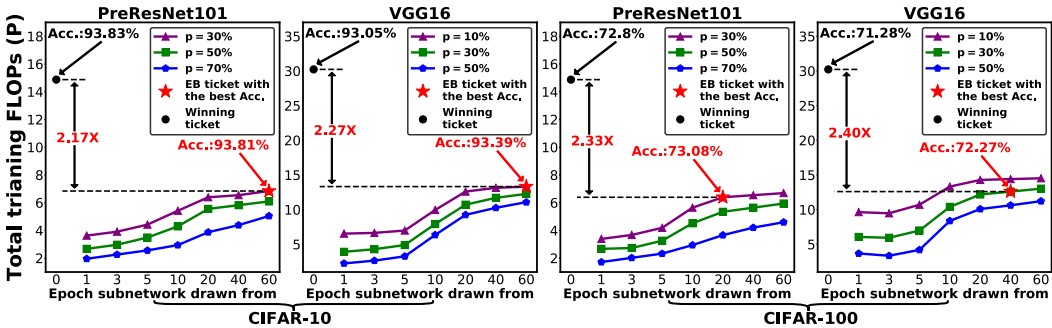

Figure 5: The total training FLOPs vs. the epochs at which the subnetworks are drawn from, for both the PreResNet101 and VGG16 models on the CIFAR-10 and CIFAR-100 datasets, where $p$ indicates the channel-wise pruning ratio for extracting the subnetworks. Note that the EB tickets at all cases achieve comparable or higher accuracies and consume less FLOPs than those of the "ground-truth" winning tickets (drawn after the full training of 160 epochs).

## 4.2 How to Implement EB Train?

From Figure 1, we can see that the EB Train scheme consists of a training (searching) step to identify EB tickets and a retraining step to retrain the EB tickets for achieving the target accuracy. *Algorithm* 1 describes the step of searching for EB tickets. Specifically, the EB searching process 1) first initializes the weights $W$ and scaling factors $\mathbf{r}$; 2) iterates the structured pruning process as in (Liu et al. (2017)) to calculate the mask distances between the consecutive resulting subnetworks, storing them into a first-in-first-out (FIFO) queue with a length of $l = 5$; and 3) exits when the maximum mask distance in the FIFO is smaller than a specified threshold $\epsilon$ (default 0.1 with the normalized distances of [0,1]). The output is the resulting EB ticket denoted as $f(x; m_{t^*} \odot W)$, which will be retrained further to reach the target accuracy. Note that unlike the lottery ticket training in (Frankle & Carbin, 2019), EB Train inherits the unpruned weights from the drawn EB ticket instead of rewinding to the original initialization, as it has been shown that deeper networks are not robust to reinitializion with untrained weights (Frankle et al., 2020).

## 4.3 How does EB Train Perform Compared to State-of-the-art Techniques?

**Experiment Setting.** We consider training the VGG16 and PreResNet101 models on both CIFAR-10/100 and ImageNet datasets following the basic setting of (Liu et al., 2018b) (i.e., training 160 epochs for CIFAR and 90 epochs for ImageNet). We measure the training *energy costs from real-device operations* as the energy cost consists of both computational and data movement costs, the latter of which is often dominant but can not captured by the commonly used metrics, such as the number of FLOPs (Chen et al., 2017), we evaluate the proposed techniques against the baselines in terms of accuracy and real measured energy consumption. Specifically, all the energy consumption of full-precision models are obtained by training the corresponding models in an embedded GPU (NVIDIA JETSON TX2). The GPU measurement setup can be found in the Appendix. Note that the energy measurement results include the overhead of using the mask distance to detect the emergence of EB tickets, which is found negligible as compared to the total training cost. For example, for the VGG16 model, the overhead caused by computing mask distances is $\leq 0.029\%$ in memory storage size, $\leq 0.026\%$ in FLOPs, and $\leq 0.07\%$ in real-measured energy costs. This is because 1) the mask distance evaluation only involves simple hamming distance calculations and 2) each epoch only calculates the distance once.

**Results and Analysis.** We first compare the computational savings with the baselines using a *progressive pruning and training* scheme (Liu et al., 2017) in terms of the total training FLOPs, when drawing subnetworks at different epochs. Figure 5 summarizes the results of the PreRes-Net101/VGG16 models and CIFAR-10/100 datasets, corresponding to the same set of experiments as in Figure 1. We see that EB Train can achieve $\mathbf{2.2 \sim 2.4\times}$ **FLOPs reduction** over the baselines, while leading to comparable or even better accuracy (- 0.81% ~ + 2.38% over the baseline).

Table 2 compares the retraining accuracy and consumed FLOPs/energy of EB Train with four state-of-the-art *progressive pruning and training* techniques: the original lottery ticket (LT) training (Frankle & Carbin, 2019), network slimming (NS) (Liu et al., 2017), ThiNet (Luo et al., 2017) and SNIP (Lee et al., 2019). While all of them involve the process of training a dense network, pruning it, and retraining, they apply different pruning criteria, e.g., NS imposes $L_1$-sparsity on channel-wise

Table 2: Comparing the accuracy and energy/FLOPs of EB Train (including its variants), NS (Liu et al. (2017)), LT (Frankle & Carbin, 2019), SNIP (Lee et al., 2019), and ThiNet (Luo et al. (2017)).

| Setting | Methods | Retrain acc. | | | Energy cost (KJ)/FLOPs (P) | | |
|---|---|---|---|---|---|---|---|
| | | p=30% | p=50% | p=70% | p=30% | p=50% | p=70% |
| PreResNet -101 CIFAR-10 | LT (one-shot) | 93.70 | 93.21 | **92.78** | 6322/14.9 | 6322/14.9 | 6322/14.9 |
| | SNIP | 93.76 | 93.31 | 92.76 | 3161/7.40 | 3161/7.40 | 3161/7.40 |
| | NS | 93.83 | 93.42 | 92.49 | 5270/13.9 | 4641/12.7 | 4211/11.0 |
| | ThiNet | 93.39 | 93.07 | 91.42 | 3579/13.2 | 2656/10.6 | 1901/8.65 |
| | EB Train (re-init) | 93.88 | 93.29 | 92.39 | 2817/7.75 | 2382/7.05 | 1565/3.77 |
| | EB Train (FF) | **93.91** | **93.90** | 92.49 | 2370/6.50 | 1970/5.70 | 1452/3.50 |
| | EB Train (LF) | 93.48 | 93.31 | 92.24 | 2265/6.45 | 1667/5.39 | 1338/3.44 |
| | EB Train (LL) | 93.24 | 92.85 | 92.12 | **489.4/6.45** | **410.9/5.39** | **281.8/3.44** |
| | **EB Train Improv.** | **0.08** | **0.48** | -0.29 | **6.5×/1.1×** | **6.5×/1.4×** | **6.7×/2.2×** |
| VGG16 CIFAR-10 | LT (one-shot) | 93.18 | 93.25 | **93.28** | 746.2/30.3 | 746.2/30.3 | 746.2/30.3 |
| | SNIP | 93.20 | 92.71 | 92.30 | 373.1/15.1 | 373.1/15.1 | 373.1/15.1 |
| | NS | 93.05 | 92.96 | 92.70 | 617.1/27.4 | 590.7/25.7 | 553.8/23.8 |
| | ThiNet | 92.82 | 91.92 | 90.40 | 298.0/22.6 | 383.9/19.0 | 380.1/16.6 |
| | EB Train (re-init) | 93.11 | 93.23 | 92.71 | 290.4/14.4 | 237.3/12.0 | 200.5/9.45 |
| | EB Train (FF) | **93.39** | **93.26** | 92.71 | 256.4/**12.7** | 213.4/10.8 | 184.2/9.85 |
| | EB Train (LF) | 93.20 | 93.19 | 92.91 | 250.1/12.8 | 199.4/10.7 | 170.3/8.51 |
| | EB Train (LL) | 93.25 | 93.13 | 92.60 | **56.1**/12.8 | **43.1**/10.7 | **36.5/8.51** |
| | **EB Train Improv.** | **0.19** | **0.01** | - 0.57 | **6.6×/1.2×** | **8.6×/1.4×** | **10.2×/1.8×** |
| PreResNet -101 CIFAR-100 | LT (one-shot) | 71.90 | 71.60 | 69.95 | 6095/14.9 | 6095/14.9 | 6095/14.9 |
| | SNIP | 72.34 | 71.63 | 70.01 | 3047/7.40 | 3047/7.40 | 3047/7.40 |
| | NS | 72.80 | 71.52 | 68.46 | 4851/13.7 | 4310/12.5 | 3993/10.3 |
| | ThiNet | 73.10 | 70.92 | 67.29 | 3603/13.2 | 2642/10.6 | 1893/8.65 |
| | EB Train (re-init) | 73.23 | **73.36** | 71.05 | 2413/7.35 | 2016/6.25 | 1392/3.53 |
| | EB Train (FF) | **73.52** | 73.15 | **72.29** | 2020/**6.40** | 1769/5.45 | 1294/3.28 |
| | EB Train (LF) | 73.41 | 73.02 | 70.72 | 2038/6.42 | 1614/5.45 | 1171/2.99 |
| | EB Train (LL) | 73.04 | 71.82 | 69.45 | **434.4**/6.42 | **366.5/5.45** | **247.3/2.99** |
| | **EB Train Improv.** | **0.42** | **1.73** | **2.28** | **7.0×/1.2×** | **7.2×/1.4×** | **7.6×/2.5×** |
| | | p=10% | p=30% | p=50% | p=10% | p=30% | p=50% |
| VGG16 CIFAR-100 | LT (one-shot) | **72.62** | 71.31 | 70.96 | 741.2/30.3 | 741.2/30.3 | 741.2/30.3 |
| | SNIP | 71.55 | 70.83 | 70.35 | 370.6/15.1 | 370.6/15.1 | 370.6/15.1 |
| | NS | 71.24 | 71.28 | 69.74 | 636.5/29.3 | 592.3/27.1 | 567.8/24.0 |
| | ThiNet | 70.83 | 69.57 | 67.22 | 632.2/27.4 | 568.5/22.6 | 381.4/19.0 |
| | EB Train (re-init) | 71.65 | 71.48 | 69.66 | 345.3/16.3 | 300.0/13.7 | 246.8/10.6 |
| | EB Train (FF) | 71.81 | **72.17** | **71.28** | 287.7/14.1 | 262.2/**12.2** | 221.7/9.85 |
| | EB Train (LF) | 71.60 | 71.50 | 70.27 | 270.5/14.0 | 262.7/12.8 | 208.7/10.1 |
| | EB Train (LL) | 71.34 | 70.53 | 69.91 | **54.6**/14.0 | **64.4**/12.8 | **44.4/10.1** |
| | **EB Train Improv.** | - 0.81 | **0.86** | **0.32** | **6.8×/1.1×** | **5.8×/1.2×** | **10.7×/1.5×** |

scaling factors from BN layers, and ThiNet greedily prunes the channel that has the smallest effect on the next layer's activation values. For EB Train, we by default follow (Liu et al., 2017) to inherit the same weights when re-training the searched ticket, adopting floating points for both the search and retrain stages (i.e., EB Train FF). We also notice existing debates (Liu et al., 2018b) on the initialization re-use, and thus also compare with a variant by re-training the ticket from a new random initialization, termed as *EB Train (re-init)*. The comparisons in Table 2 further show that inheriting weights from the EB tickets favor the generalization of retraining as compared to both the random initialization and "over-cooked" weights, aligning well with the recent discussion between rewinding and fine-tuning (Renda et al., 2020). Furthermore, we apply the proposed EB Train on top of the low-precision training method (Yang et al., 2019a) and obtain experiment results of another two variants of EB Train: 1) EB Train with low-precision search and full-precision retrain (i.e., EB Train LF), and 2) EB Train with both low-precision search and retrain (i.e., EB Train LL).

Table 2 demonstrates that EB Train consistently outperforms all competitors in terms of saving training energy and computational costs, meanwhile improving the final accuracy in most cases. We use *EB Train Improv.* to record the performance margin (either accuracy or energy/computation) between EB Train and the **strongest** competitor among the four state-of-the-art baselines. Specifically, EB Train with full-precision floating point (FP32) search and retrain outperforms those pruning

Table 3: Comparing the accuracy and total training FLOPs of EB Train, Network Slimming (Liu et al., 2017), ThiNet (Luo et al., 2017), and SFP (He et al., 2018). The "Acc. Improv." is the accuracy of the pruned model minus that of the unpruned model, so a positive number means the pruned model has a higher accuracy.

| Models | Methods | Pruning ratio | Top-1 Acc. (%) | Top-1 Acc. Improv. (%) | Top-5 Acc. (%) | Top-5 Acc. Improv. (%) | Total Training FLOPs (P) | Total Training Energy (MJ) |
|---|---|---|---|---|---|---|---|---|
| ResNet18 ImageNet | Unpruned | - | 69.57 | - | 89.24 | - | 1259.13 | 98.14 |
| | NS | 10% | 69.65 | +0.08 | 89.20 | -0.04 | 2424.86 | 193.51 |
| | | 30% | 67.85 | -1.72 | 88.07 | -1.17 | 2168.89 | 180.92 |
| | SFP | 30% | 67.10 | -2.47 | 87.78 | -1.46 | 1991.94 | 158.14 |
| | **EB Train** | 10% | **69.84** | **+0.27** | **89.39** | **+0.15** | 1177.15 | 95.71 |
| | | 30% | 68.28 | -1.29 | 88.28 | -0.96 | **952.46** | **84.65** |
| ResNet50 ImageNet | Unpruned | - | 75.99 | - | 92.98 | - | 2839.96 | 280.72 |
| | ThiNet | 30% | 72.04 | -3.55 | 90.67 | -2.31 | 4358.53 | 456.13 |
| | | 50% | 71.01 | -4.58 | 90.02 | -2.96 | 3850.03 | 431.73 |
| | | 70% | 68.42 | -7.17 | 88.30 | -4.68 | 3431.48 | 416.44 |
| | **EB Train** | 30% | **73.86** | **-1.73** | **91.52** | **-1.46** | 2242.30 | 232.18 |
| | | 50% | 73.35 | -2.24 | 91.36 | -1.62 | 1718.78 | 188.18 |
| | | 70% | 70.16 | -5.43 | 89.55 | -3.43 | **890.65** | **121.15** |

methods by up to $1.2 \sim 4.7\times$ and $1.1 \sim 4.5\times$ in terms of the energy consumption and computational FLOPs, while always achieving comparable or even better accuracies, across three pruning ratios, two DNN models and two datasets. Moreover, EB Train with both low-precision (8 bits block floating point (FP8)) search and retrain outperforms the baselines by up to $5.8 \sim 24.6\times$ and $1.1 \sim 5.0\times$ in terms of energy consumption estimated using real-measured unit energy from (Yang et al., 2019b) and computational FLOPs. Note that FP8 and FP32 are counted as the same FLOPs count units (Sohoni et al., 2019). In addition, comparing with the *re-init* variant endorses the effectiveness of initialization inheritance in EB Train. As an additional highlight, EB Train naturally leads to more efficient inference of the pruned DNN models, unifying the boost of efficiency throughout the full learning lifecycle.

**ResNet18/50 on ImageNet.** To study the performance of EB Train in a harder dataset, we conduct experiments using ResNet18/50 and ImageNet, and compare its resulting accuracy, total training FLOPs and energy cost with those of the method in (He et al., 2016a), NS (Liu et al., 2017), ThiNet (Luo et al., 2017) and SFP (He et al., 2018) as summarized in Table 3. We have **three observations**: 1) When training ResNet18 on ImageNet, EB Train achieves a better accuracy (+0.27%) over the unpruned one while introducing 10% channel sparsity; 2) EB Train outperforms other baselines for both ResNet18 and ResNet50 by reducing the total training costs while achieving a comparable or better accuracy. Specifically, EB Train achieves a reduced training FLOPs of 51.5% ~ 56.1% and 48.6% ~ 74.0% and a reduced training energy of 46.5% ~ 53.2% and 49.1% ~ 70.9%, while leading to a better top-1 accuracy of +0.19% ~ +1.18% and +1.74% ~ +2.34% for ResNet18 and ResNet50, respectively. For example, when training ResNet50 on ImageNet, EB Train with a pruning ratio of 70% achieves better (+1.74%) accuracy over ThiNet while saving 74% total training FLOPs and 71% training energy; 3) Different from the results on CIFAR-10/100 shown in Table 2, all methods performed on ImageNet (a harder dataset) start to yield accuracy reductions (-1.72% ~ -3.55%) for ResNet18/50 with a pruning ratio of only 30%, compared with the unpruned one. Note that in Table 3, the unpruned results are based on the official implementation in (He et al., 2016a); The SFP results are obtained from their original paper (He et al., 2018), which does not provide results at a pruning ratio of 10%; all the ThiNet results under various pruning ratios are obtained from the original paper (Luo et al., 2017); and the NS results are obtained by conducting the experiments ourselves.

## 5 DISCUSSION AND CONCLUSION

We have demonstrated that winning tickets can be drawn at the very early training stage, i.e., EB tickets exist, in both the standard training and several lower-cost training schemes. That motivates a practical success of applying EB tickets to efficient training, whose results compare favorably against state-of-the-arts. Moreover, experiments show that EB Train with low-precision search and retraining achieve more efficient training. We believe many promising problems still remain open to be addressed. An immediate future work is to test low-precision EB Train algorithms on larger models/datasets. We are also curious whether more lower-cost training techniques could be associated with EB Train. Finally, sometimes high pruning ratios (e.g, $p \geq 0.7$) can hurt the quality of EB tickets and the retrained networks. We look forward to automating the choice of $p$ for different models/datasets, unleashing higher efficiency.

## 6 ACKNOWLEDGEMENT

The work is supported by the National Science Foundation (NSF) through the Real-Time Machine Learning (RTML) program (Award number: 1937592, 1937588).

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

# A APPENDIX

## A.1 MEASUREMENT OF ENERGY COST

Figure 6 shows our GPU measurement setup, in which the GPU board is connected to a laptop and a power meter. In particular, the training settings are downloaded from the laptop to the GPU board, and the real-measured energy consumption is obtained via the power meter and runtime measurement for the whole training process.

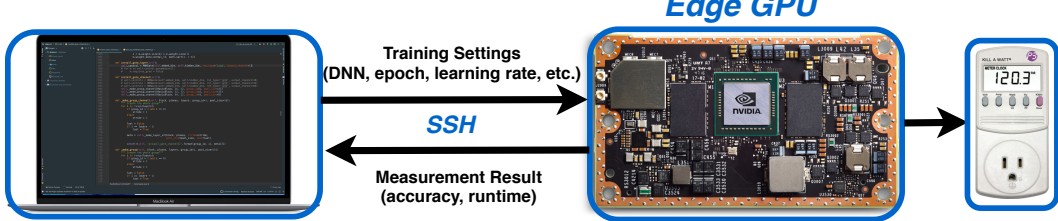

Figure 6: The energy measurement setup (from left to right) with a MAC Air latptop, an embedded GPU (JETSON TX2 (NVIDIA Inc.)), and a power meter.

## A.2 APPROPRIATE LARGE LEARNING RATE FAVORS THE EMERGENCE OF EB TICKETS

Here we show the effect of larger learning rates setting. The original schedule is denoted as $[80_{LR \to 0.01}, 120_{LR \to 0.001}]$, i.e., starting from 0.1, decayed to 0.01 at the 80th epoch, and further decayed to 0.001 at the 120th epoch. In comparison, we test a larger initial learning rate, staring from 0.5 and 0.2 for VGG16 and PreResNet101 datasets, respectively. After drawing tickets, we retrain them all using the same learning rate schedule with their searching phase for sufficient training and a fair comparison. We see from Table 4 that high-quality EB tickets also emerge early when searching with the larger initial learning rate, whose final accuracies are even better.

Table 4: The retraining accuracy of subnetworks drawn at different training epochs using different inital learning rate, where the pruning ratio is 0.5.

| | LR Initial | Retrain acc.(%) (CIFAR-100) | | | | Retrain acc.(%) (CIFAR-10) | | | |
|---|---|---|---|---|---|---|---|---|---|
| | | 10 | 20 | 40 | final | 10 | 20 | 40 | final |
| VGG16 | 0.1 | **71.11** | 71.07 | 69.14 | 69.74 | 93.26 | 93.34 | 93.20 | 92.96 |
| | 0.5 | 70.69 | **71.65** | **71.94** | **71.58** | **93.49** | **93.45** | **93.44** | **93.29** |
| PreResNet101 | 0.1 | 71.58 | 72.67 | 72.67 | 71.52 | **93.60** | 93.46 | 93.56 | 93.42 |
| | 0.2 | **72.58** | **72.90** | **72.86** | **72.71** | 93.40 | **93.46** | **93.87** | **93.69** |

