# OpenReview forum: "Drawing Early-Bird Tickets: Toward More Efficient Training of Deep Networks"
_ICLR.cc/2020/Conference — Accept (Spotlight)_

### Official Review · AnonReviewer2 · 2019-10-16
**Official Blind Review #2**

**Rating:** 6

**Review:**

This paper presents a method to speed up training of deep neural networks. The main contribution is a method to quickly identify winning lottery tickets (denoted early-bird, or EB by the authors), without running the model to convergence. The authors present interesting preliminary experiments that motivate their method, and show that it works on two image recognition datasets using two models.

This paper addresses an under-explored, but very important problem in AI: the increasing cost of training models. The authors present interesting evidence about the potential to detect EBs early on. The experiments presented in Figures 1 and 3 are convincing and will be of interest to the community. The proposed method seems to work, at least on the setups explored by the authors. I am leaning towards acceptance, but am concerned with the following:

1. The authors experiment with a limited set of datasets (CIFAR-10 is a relatively easy task), and with a set of non-competitive baselines (SOTA for CIFAR-10/100 is 99%/91.3%, see https://benchmarks.ai/cifar-10{,0}). I would have liked to see whether the proposed method translates to harder datasets and stronger models.

2. I might be missing something here, but to the best of my understanding the large learning rate part (page 4) does not demonstrate the benefits of increasing the learning rate, but the problems with *decreasing* it. The two might seem like the same thing, but in fact they're not: the authors claim the [80,120] policy is standard, and use it when training the subnetwork, so showing that [0,100] is inferior does not present a way to improve over the current approach, but evidence that the other approaches are inferior.

Other questions:
1. In Figure 1, it seems that the extracted subnetworks are doing very well even after 0 epochs. Does this mean that a trained version of a random subnetwork could reach within 1-2 points of the unpruned model? or is it pruned after training for 1 epoch?

2. If I understand correctly, Figure 5 should be illustrating the proposed method, which automatically identifies the early stopping point. In that case, I am not sure why the plot is a function of the epoch.

3. Do the authors have any intuition as to the sharp decrease in the 70% graph in Figures 1 and 2 around epoch 50?

Writing:

1. The language used by the authors is sometimes exaggerated. Expressions such as "bold guess" (section 3.2), "innovative ... scheme" (section 4) and comparisons to Winston Churchill would be better left out of the manuscript.

2. Typos and such:
-- several across the paper. For instance:
- Intro: After *bring* identified (should be "being")
- Related work: when training *it* isolation (in)

-- Missing venue for Frankle and Corbin (2019)


**Experience Assessment:**

I have published one or two papers in this area.

**Review Assessment: Checking Correctness Of Derivations And Theory:**

N/A

**Review Assessment: Checking Correctness Of Experiments:**

I assessed the sensibility of the experiments.

**Review Assessment: Thoroughness In Paper Reading:**

I read the paper thoroughly.

---

> ### Author Response · Authors · 2019-11-13
> **Response to Reviewer#2**
>
> We thank the reviewer for the insightful comments. We have revised the manuscript and added the following response to address your concerns and questions:
>
> C1: ResNet and VGG on CIFAR-10/100 are popular benchmarks widely used in both latest “lottery ticket” and efficient CNN training papers. Furthermore as requested, we add a group of experiments on ImageNet and ResNet18 show that our method translates to a harder dataset. As shown in the following table, we compare the retrain accuracy and training FLOPs of EB Train with those of the unpruned ResNet18, network slimming, and SFP on ResNet18 + ImageNet.
>
> ---------------------------------------------------------------------------------------------------------------------------------------------
> Methods    Pruning ratio   Top-1 Acc.   Acc. Improv. (%)   Top-5 Acc.   Acc. Improv. (%)   Training FLOPs (P)
> ---------------------------------------------------------------------------------------------------------------------------------------------
> Unpruned      	 -	          69.57            	-	                89.24	             -	                         1259.13
> NS	                       10%	          69.65                +0.08	                89.20	         -0.04	                 2424.86
> NS	                       30%	          67.85	            -1.72	                88.07	         -1.17        	         2168.89
> SFP	                       30%	          67.10	            -2.47	                87.78	         -1.46		         1991.94
> EB-Train	               10%	          69.84	           +0.27	                89.39               +0.15		         1177.15
> EB-Train	               30%	          68.28	            -1.29	                88.28	         -0.96		         952.46
> ---------------------------------------------------------------------------------------------------------------------------------------------
>
> The above table shows that 1) when the pruning ratio is 10%, EB Train achieves a better accuracy (+0.27 vs. +0.08 over the unpruned one) while requiring 51% less total training FLOPs, as compared to the network slimming (NS) pruning method; 2) when the pruning ratio is 30%, EB Train results in a higher accuracy (-1.29 vs. -1.72(NS)/-2.47(SFP) over the unpruned one) while reducing the training FLOPs by 56% and 52%, as compared to the NS and SFP baselines. Note that “Acc. Improv.” in the above table is referred to that of the unpruned models. In the above table, the unpruned results are based on the official implementation (see https://github.com/facebookarchive/fb.resnet.torch; The SFP results are obtained from their original paper (see https://arxiv.org/pdf/1808.06866.pdf), which did not provide results at 10% pruning ratio; and the NS results are obtained by conducting the experiments ourselves.
>
> C2: Thank you for pointing out this and we agree with your comment. Also, we have conducted experiments to show the retrain accuracy of sub-networks drawn from different learning rate schedules. Please kindly see our response to Q2 of Reviewer 1.
>
> Q1: It is pruned after training for 1 epoch because Figure 1 plots the retrain accuracy of the sub-networks obtained by pruning after training for M epochs (M starts from 1). Sorry for not having made it clear enough.
> While sometimes the extracted networks after training for 1- 2 epochs might sometimes achieve good accuracy, they can be unstable. As an illustration, we have plotted error bars for all data points in Figures 1 and 2 based on three independent runs.
>
> Q2: Figure 5 shows the total training FLOPs of EB Train vs. different early-stop epochs as an “ablation” example to show the effectiveness of EB Train. We apologize for the confusion. Meanwhile, we use Table 2 to illustrate the proposed mask distance-based method which automatically identifies the early stopping point, by benchmarking over the state-of-the-art designs.
>
> Q3: Please kindly find our response in the reply to Q1 of Reviewer 1.
>
> Thank you for providing your kind suggestions on our writing! We have: 1) toned down our language in the revised manuscript; and 2) corrected the typos. We will carefully proofread our paper.

---

> > ### Comment · AnonReviewer2 · 2019-11-14
> > **Response to your comments**
> >
> > Thank you for the clarifications.
> >
> > 1. How come the FLOP numbers of the pruned NS and SFP models are so much higher than the unpruned baseline?
> >
> > 2. To the best of my understanding, the table shows that indeed, using a stronger baseline and a harder datasets yields a slightly different picture: compared to Table 2, which shows that pruning up to 70% yields similar or better performance compared to the baseline, here even 30% pruning reduces performance, and I can only imagine that 50% and 70% would yield even larger reductions. I would encourage the authors to add these results to the paper and discuss them.

---

> > > ### Author Response · Authors · 2019-11-15
> > > **Response to Reviewer#2**
> > >
> > > Thanks for your careful review and feedback.
> > >
> > > Q1:
> > > Sorry for the confusion. The FLOPs of all the pruned models (including EB Train) in the table consist of both winning ticket search and retraining costs, hence leading to higher FLOPs of the pruned NS and SFP models over the unpruned model.
> > >
> > > Q2:
> > > We appreciate your suggestion and share your curiosity. Due to the limited time frame in rebuttal, we actually did not have enough time and resources to finish the experiments under 50% and 70% pruning.
> > >
> > > We will continue our experiments and make sure to obtain those results to be updated in the final version, to verify your "imagination". We would also try the low-precision schemes (e.g., EB Train LL) and see if better accuracy-training FLOPs trade-offs could be achieved.

---

### Official Review · AnonReviewer3 · 2019-10-22
**Official Blind Review #3**

**Rating:** 6

**Review:**

The authors further study the lottery ticket hypothesis formulated by Frankle and Carbin. They demonstrate that the sparsity pattern corresponding to a lottery ticket for a given initialization can be uncovered via low-cost training. By doing so, they propose a method to: 1) first identify the lottery ticket efficiently and 2) exploit the sparsity of the resulting network to train it at a lower cost.

The contribution is however a bit incremental in my opinion. The original LT paper was not focused on efficiency, and it is not a far stretch to try to find the tickets sooner during the training. On the other hand, the experiments are well conducted (especially 4.3) and even if the original idea is simple, it is of interest to see it tested as clearly. All in all, I found this paper convincing and worth reading, and I think it should be accepted.

Positive points:
- The literature review is sufficient and present with great clarity the latest results.
- The problem tackled is of great interest and has potentially impactful applications.
- The authors focus on hardware friendly types of pruning.
- The paper is well written and enjoyable.
- The algorithm used to compute the EB tickets seems a bit ad hoc, but in my opinion sufficient as a first approach.


Nitpicking:
- Not sure why Max(Q) > eps is in the while condition (return if Max(Q) < eps should be sufficient)
- The treatment of the mask distance in figure 3 is confusing. It is not obvious why the authors are plotting 1-distance, and the legend of the figure suggest that a mask has a distance of 1 with itself. Recommend plotting d instead of 1-d and invert the color bar instead if they feel so inclined (yellow=0).
- Abstract: “consistently exist across models and datasets” a bit of a strong claim as only cifar is used.


**Experience Assessment:**

I have read many papers in this area.

**Review Assessment: Checking Correctness Of Derivations And Theory:**

I assessed the sensibility of the derivations and theory.

**Review Assessment: Checking Correctness Of Experiments:**

I assessed the sensibility of the experiments.

**Review Assessment: Thoroughness In Paper Reading:**

I read the paper thoroughly.

---

> ### Author Response · Authors · 2019-11-13
> **Response to Reviewer#3**
>
> Thanks for your careful review and positive feedback! We have addressed your comments in our revised manuscript as summarized below:
>
> we have followed your kind suggestions to 1) remove the while condition (Max(Q) > eps) in algorithm 1 for making it more concise; and 2) plot d and invert the color bar in Figure 3 to better visualize the effectiveness of the proposed mask distance.
>
> We have toned down the strong statement in the abstract. Furthermore, we have added experiments on the ImageNet dataset in Table 3 to more thoroughly validate our claim.

---

> > ### Comment · AnonReviewer3 · 2019-11-14
> > **Response to the Authors**
> >
> > Thank you for addressing my comments. The experiments on ImageNet are a nice addition too.

---

### Official Review · AnonReviewer1 · 2019-10-23
**Official Blind Review #1**

**Rating:** 8

**Review:**

The paper empirically analyzed the wide existence of "early-bird tickets", e.g., the "lottery tickets" emerging and stabilizing in very early training stage. The potential connection to (Achille et al., 2019; Li et al.,2019) reads interesting.

The authors made several contributions in addition to the observation: (1) the EB tickets stay robust under large learning rates (while early stopping) and low-precision training; (2) the EB tickets can be detected using epoch-wise consecutive comparison (mask distance), rather than comparing with some oracle ticket; (3) the application of EB ticket towards energy efficient training, which is interesting as this is perhaps the first practical application demonstrated of lottery ticket.

While I like how the paper connects theory hypothesis to real applications, the experiments need to be solidified in a few aspects:

1) Figures 1 and 2, why a few plunges of curves (say Fig 2.a, p = 70%)? Does it imply the training might not be stable?

2) Table 1, the authors test two lr schedules to show "large learning rates favor the emergence of EB Tickets". Yet the choice of lr matters a lot and can be tricky. Why the authors pick the two specific learning rate schedules? Why are they "comparable"? What if being more aggressive in choosing larger lr, say starting from lr= 0.5?

3) The low precision EB ticket is not actually applied or evaluated in Section 4. It would have been interesting to see.

4) I fail to find the 4.7 times energy saving as claimed in abstract from Table 2?


**Experience Assessment:**

I have read many papers in this area.

**Review Assessment: Checking Correctness Of Derivations And Theory:**

I carefully checked the derivations and theory.

**Review Assessment: Checking Correctness Of Experiments:**

I carefully checked the experiments.

**Review Assessment: Thoroughness In Paper Reading:**

I read the paper thoroughly.

---

> ### Author Response · Authors · 2019-11-13
> **Response to Reviewer#1**
>
> Thanks for your careful review and comments and for appreciating our contributions! We have conducted more experiments and revised our paper to 1) address your comments and 2) improve the paper. Please find our itemized responses below.
>
> Q1: After the submission, we carefully revisited all experiments and found that the plunges were caused by the resulting over-pruning of the networks’ certain layers when the target pruning ratio is high (e.g., 70%), due to the adopted global pruning scheme, i.e., enforcing to reach the target compression ratio in a network-wise instead of layer-wise manner. For example, pruning VGG-16 on CIFAR100 by 70% will lead to several layers of the resulting network to have less than 10 channels.
>
> Furthermore, we found that this “plunge” issue can be fixed by applying a simple “protective pruning” heuristic on top of our current algorithm. Specifically, we stop pruning a layer, when its remaining channels after pruning become less than 10% of the original, for avoiding overly slimming this layer; meanwhile we (uniformly) prune other layers more to meet the overall pruning ratio. Experiments show that such a simple protective strategy can effectively eliminate the plunges as shown in the updated Figures 1-2, without incurring overhead. Note that this “protective pruning” is only activated when there are over-pruned layers and thus affects only a few data points in the figures.
>
> Q2: we followed the common learning rate setting for drawing winning tickets (see Section 6 of https://arxiv.org/pdf/1810.05270.pdf). As you kindly suggested, here we show the retrain accuracy of sub-networks drawn from different learning rate schedules.
>
> Interestingly, a large initial lr of 0.5 and degraded schedule [80_{LR\rightarrow0.05}, 120_{LR\rightarrow0.005}] indeed works better for drawing EB tickets in VGG16 performed on CIFAR-10/100, leading to an earlier emergence of EB tickets and a higher retrain accuracy. This seems promising and is consistent with our hypothesis “large learning rates favor the emergence of EB Tickets.”
>
> (VGG16@CIFAR10)
> ---------------------------------------------
> Initial LR   Epoch EB drawn from
> 		  10       20       40	 final
> ---------------------------------------------
>      0.1     93.26  93.34  93.20  92.96
>      0.5     93.49  93.45  93.44  93.29
> ---------------------------------------------
> (VGG16@CIFAR100)
> ---------------------------------------------
> Initial LR   Epoch EB drawn from
> 		  10       20       40	 final
> ---------------------------------------------
>      0.1     71.11  71.07  69.14  69.74
>      0.5     70.69  71.65  71.94  71.58
> ---------------------------------------------
>
> A similar observation is found when the initial lr is 0.2, in experiments on ResNet + CIFAR-10/100, although the schedule of initial lr 0.5 does not seem to further help EB tickets:
>
> (PreResNet101@CIFAR10)
> ---------------------------------------------
> Initial LR   Epoch EB drawn from
> 		  10       20       40	 final
> ---------------------------------------------
>      0.1    93.60  93.46  93.56  92.42
>      0.2    93.40  93.46  93.87  93.69
> ---------------------------------------------
>
> (PreResNet101@CIFAR100)
> ---------------------------------------------
> Initial LR   Epoch EB drawn from
> 		  10       20       40	 final
> ---------------------------------------------
>      0.1    71.58  72.67  72.67  71.52
>      0.2    72.58  72.90  72.86  72.71
> ---------------------------------------------
>
> Given the aforementioned observation in experiments with various models and datasets (thanks to the Q2 comment from review#2), we revise the claim in the paper to a more accurate one as “appropriate large learning rates can enable an earlier emergence of EB tickets”. We will conduct more experiments with even larger lrs to find more insights on “lr vs. emergence time of EB tickets” and update in the camera ready version.
>
> Q3: Indeed, applying low-precision EB tickets on top of low/full precision retraining is more interesting as it can achieve more energy savings while maintaining a comparable retrain accuracy. As you kindly suggested, we add two sets of corresponding experiment results into Table 2: 1) EB Train with low precision search and full precision retrain (EB-Train LF) and 2) EB Train with both low precision search and retrain. Specifically, the resulting energy savings and training FLOPs are 5.8-24.6x and 1.1-5.0x over the baseline competitors.
>
> Q4: the 4.7x energy savings can be found in Table 2’s experiment on PreResNet-101@CIFAR100: when the pruning ratio is 70%, the energy savings of EB-Train compared to the lottery ticket (one-shot) baseline is 6095/1294 (\approx 4.7x). Note that additional experiments as responding to your Q3 comment show that EB Train (with both low precision search and retrain) can lead up to 24.6x energy savings.

---

### Decision · Program_Chairs · 2019-12-19

**Decision:**

Accept (Spotlight)

**Comment:**

This work studies small but critical subnetworks, called winning tickets, that have very similar performance to an entire network, even with much less training. They show how to identify these early in the training of the entire network, saving computation and time in identifying them and then overall for the prediction task as a whole.

The reviewers agree this paper is well-presented and of general interest to the community. Therefore, we recommend that the paper be accepted.